# Antioxidant Capacity, Vitamin C and Polyphenol Profile Evaluation of a *Capsicum chinense* By-Product Extract Obtained by Ultrasound Using Eutectic Solvent

**DOI:** 10.3390/plants11152060

**Published:** 2022-08-06

**Authors:** Kevin Alejandro Avilés-Betanzos, Julio Enrique Oney-Montalvo, Juan Valerio Cauich-Rodríguez, Marisela González-Ávila, Matteo Scampicchio, Ksenia Morozova, Manuel Octavio Ramírez-Sucre, Ingrid Mayanin Rodríguez-Buenfil

**Affiliations:** 1Centro de Investigación y Asistencia en Tecnología y Diseño del Estado de Jalisco A.C., Subsede Sureste, Tablaje Catastral 31264, Km. 5.5 Carretera Sierra Papacal-Chuburná Puerto, Parque Científico Tecnológico de Yucatán, Sierra Papacal 97302, Mexico; 2Centro de Investigación Científica de Yucatán, Unidad de Materiales, Calle 43 No. 130 x 32 y 34, Colonia Chuburná de Hidalgo, Mérida 97205, Mexico; 3Centro de Investigación y Asistencia en Tecnología y Diseño del Estado de Jalisco, Ex vivo Digestion Laboratory, CIATEJ, Normalistas No. 800, Colinas de la Normal 44270, Mexico; 4Faculty of Science and Technology, Free University of Bolzen-Bolzano, Piazza Università 1, 39100 Bolzano, Italy

**Keywords:** *Capsicum chinense*, by-product, antioxidant capacity, polyphenols, natural deep eutectic solvent, NADES

## Abstract

Habanero pepper leaves and stems (by-products) have been traditionally considered waste; however, bioactive compounds such as polyphenols, vitamin C and carotenoids have been identified that can be used for formulation of nutraceuticals or functional foods. Furthermore, the extraction of these bioactive compounds by using environmentally friendly methods and solvents is desirable. Thus, the aim of this study was to assess the antioxidant capacity, total polyphenol content (TPC), the phenolic profile and vitamin C content in extracts obtained from by-products (stems and leaves) of two varieties (Mayapan and Jaguar) of habanero pepper by ultrasound-assisted extraction (UAE) using natural deep eutectic solvents (NADES). The results showed that NADES leads to extracts with significantly higher TPC, higher concentrations of individual polyphenols (gallic acid, protocatechuic acid, chlorogenic acid, cinnamic acid, coumaric acid), vitamin C and, finally, higher antioxidant capacity (9.55 ± 0.02 eq mg Trolox/g DM) than UAE extraction performed with methanol as the solvent. The association of individual polyphenols with NADES was confirmed by principal component analysis (PCA). Overall, NADES is an innovative and promising “green” extraction technique that can be applied successfully for the extraction of phenolic compounds from habanero pepper by-products.

## 1. Introduction

The habanero pepper plant is recognized worldwide for its unique organoleptic characteristics, its shelf life and for the higher bioactive compounds content in the fruit, such as capsaicin, polyphenols, vitamin C and carotenoids, derived from a long process of adaptation to the edaphological and climatological conditions of the Yucatan peninsula (Yucatán, Quintana Roo and Campeche). Since it has been recognized that the region of growth is responsible for conveying such particular characteristics, the Habanero pepper from the Mayapan and Jaguar varieties received, in 2010, the denomination of origin [1,2].

The annual production of habanero peppers from the Yucatan Peninsula is constantly growing. Nowadays, it contributes 30% of the national production, followed by the state of Sinaloa, according to an Agri-Food and Fishing Information Service (SIAP) 2020 report. Unfortunately, the growing demand for habanero peppers also triggers an increase in waste, such as the peduncles, stems and leaves of the plant. Generally, such by-products are simply discarded (peduncles) or burned (stems and leaves) once the fruit has been harvested [3,4].

Although stems and leaves of the habanero pepper are generally considered as waste, some authors [5,6] have reported that the leaves have a high content of bioactive compounds of interest such as polyphenols, while mainly phenolic acids have been found in stems [7,8]. Some of these works have shown that such metabolites are conventionally extracted using organic solvents (methanol, ethanol, acetone, etc.) together with different extraction techniques such as ultrasound-assisted extraction or microwave extraction. However, using these solvents increases costs as well as polluting waste that affects the environment (soil, water and air) and may also become a hazard for the operators [9]. Therefore, it is necessary to implement solvents that are inexpensive, easy to handle, reusable, biodegradable, low toxic and, thus, driven by the principles of green chemistry [10].

For this purpose, in recent years, there has been a growing interest around Natural Deep Eutectic Solvents (NADES). Such solvents are recognized as GRAS (generally recognized as safe) due to their low toxicity [11], easy biodegradability and recyclability. NADES are derived from a combination of two or more natural compounds, which, through hydrogen bonds’ interaction, develop a liquid phase, normally translucent, with high viscosity and a lower melting point than the individual compounds [12]. Typical examples of NADES are maple syrup, beet sugar, intracellular fluid (sugars and amino acids) and honey [13]. In particular, honey is of great interest because of its high nutritional content of glucose, fructose, bioactive compounds, minerals and vitamins. Not surprisingly, the use of honey as a solvent has been reported in several ancient cultures, i.e., for the preparation of traditional medicines or even, recently, for the extraction of daizin, a phenolic compound precursor of daidzein that has estrogenic activity [14,15].

However, the use of NADES as the bioactive compounds’ extraction solvent is relatively recent [15]. Only a few works have been reported on the use of NADES (and honey) for the extraction of agricultural by-products [16]. A few examples include the extraction of flavonoids, alkaloids and/or total polyphenols from *Pueraria candollei* var. *mirifica* root [15], olive seed [17], wheat husk [18], *Prunus cerasus* skin [19], *Moringa oleifera* L. leaves [20], *Ginkgo biloba* leaves [21] and *Peumus boldus* leaves [22]. Generally, the results of extraction were comparable to those obtained by ultrasound-assisted extraction (UAE) with organic solvents (methanol or ethanol) [23].

Despite such few examples, there is no information on the use of NADES with UAE to extract phenolic compounds and high antioxidant capacity from by-products of *Capsicum chinense*. Instead, because of the growing importance of habanero pepper cultivation and because of the increasing production of by-products generated from its manufacture, it is of great importance to understand whether it is possible to extract valuable compounds from such by-products by using environmentally friendly technologies. Accordingly, this work aims to assess the antioxidant capacity, total polyphenol content (TPC), the phenolic profile and vitamin C content in extracts obtained from stems and leaves (by-products) of two varieties (Mayapan and Jaguar) of habanero pepper by ultrasound-assisted extraction (UAE) using natural deep eutectic solvents (NADES).

Overall, this work attempts to propose new environmentally friendly technologies that could be useful for the valorization of vegetable by-products, i.e., for the extraction of valuable sources of polyphenols and vitamins that could be used as functional ingredients and food supplements [15].

## 2. Results

### 2.1. Total Polyphenol Content and Antioxidant Capacity of By-Products’ Extracts from Capsicum chinense Jacq

Table 1 shows the total polyphenol content (TPC) and antioxidant capacity (DPPH and ABTS) of the resulting extracts obtained by three independent technologies (ultrasound assisted extraction (UAE), NADES based on choline chloride:glucose and NADES based on honey) from leaves or stems of two distinct varieties (Mayapan and Jaguar) of habanero pepper by-products.

The results showed that the highest concentration of total polyphenols (63.80 ± 0.00 mg GAE/100 g DM) was obtained with UAE applied on the Jaguar habanero leaves.

The extract from the Jaguar variety of habanero pepper leaf obtained with a natural deep eutectic solvent made of choline chloride and glucose (NADES-cl) achieved the highest concentration of polyphenols (22.38 ± 0.07 mg GAE/100 g DM) among the NADES-cl extracts. A similar behavior was observed with the extract obtained with honey at 60% (*v*/*v*) (NADES-h) from the same habanero pepper by-product and variety (39.90 ± 0.67 mg GAE/100 g DM) when compared with the rest of the NADES-h extracts.

On the other hand, apparently, the stem from the habanero pepper Mayapan variety has a low content of polyphenols, since all the extracts obtained with this variety exhibited the lowest content of polyphenols, independently of the solvent used, during ultrasound-assisted extraction: NADES-cl (9.18 ± 0.29 mg GAE/100 g DM), NADES-h (18.67 ± 0.03 mg GAE/100 g DM) and methanol (19.37 ± 0.03 mg GAE/100 g DM).

Antioxidant capacity as determined by DPPH (2,2-diphenyl-1-picrylhydrazyl) methodology shows a higher percentage of inhibition when the extracts were obtained with MeOH. The extract that presented the highest antioxidant capacity (82.83% inhibition) was obtained from Mayapan stem (Table 1); however, no statistically significant difference was shown between extracts obtained with the same solvent (MeOH) from leaf of Mayapan (78.32% inhibition) and stem (81.36% inhibition) and leaf (77.84% inhibition) of Jaguar variety. The lowest percentage of inhibition was presented by the extract acquired with NADES-h from Mayapan’s leaf (7.09% inhibition).

The DPPH inhibition percentage was higher in the extracts obtained from the stem compared to the leaves, regardless of the variety of habanero pepper or type of solvent.

By the ABTS methodology, the highest antioxidant capacity was obtained from the stem’s extract of the Mayapan variety using NADES-cl with 9.55 ± 0.02 eq mg Trolox/g DM, followed by the extract obtained with NADES-h, with 8.75 ± 0.01 eq mg Trolox/g DM, from the same by-product and variety (Table 1). The lowest antioxidant capacity was observed in the extracts obtained by MeOH. The extract derived from the leaf of the Jaguar variety habanero pepper obtained with MeOH also presented the lowest antioxidant capacity of the entire experimental design (1.58 ± 0.01 eq mg Trolox/g DM).

Figure 1a shows the significant effect of all the factors and interactions (except triple interaction) on the total polyphenol content; however, the solvent presented a major statistical effect. The Pareto chart showed that two main factors (both by-products and solvent including both NADES and MeOH) had a significant effect (*p* < 0.05) over the antioxidant capacity as determined by DPPH methodology (Figure 1b). Figure 1c depicts that the solvent and the by-product factors had a significant effect (*p* < 0.05) on the antioxidant capacity by the ABTS method, being higher with the first. The effect of all factors on TPC and antioxidant capacity is confirmed by the interaction chart (Appendix A) and principal factors chart (Appendix A).

### 2.2. Capsicum chinense By-Products Polyphenol Profile

Polyphenol profile results from a 2 × 2 × 3 experimental design are shown in Table 2. Extracts obtained from Jaguar variety stem using NADES-h as a solvent presented a higher concentration of protocatechuic acid (59.63 ± 0.13 mg/100 g MS), chlorogenic acid (30.29 ± 0.34 mg/100 g MS) and catechin (69.88 ± 1.16 mg/100 g MS) compared to the other extracts.

On the other hand, the extracts obtained from Jaguar leaves using NADES-h as a solvent showed the highest concentrations of coumaric acid (2.77 ± 0.07 mg/100 g DM), cinnamic acid (7.20 ± 0.41 mg/100 g DM), vanillin (5.31 ± 0.19 mg/100 g DM), neohesperidin (10.94 ± 8.17 mg/100 g DM) and naringenin (2.06 ± 0.01 mg/100 g DM), with a statistically significant difference according to one-way ANOVA (*p* < 0.05). These data can be confirmed in Appendix A chromatogram.

It was shown that the extracts obtained from the Mayapan variety habanero pepper leaf through the use of NADES-h presented the highest concentration of rutin and quercetin + luteolin, with 28.62 ± 0.58 mg/100 g DM and 52.68 ± 2.51 mg/100 g DM, respectively, in comparison with the other extracts. Finally, gallic acid was found in a higher concentration in the extract obtained by UAE from the Mayapan variety leaf by-product, using NADES-cl.

The extract recovery from habanero pepper Jaguar leaf with methanol (Appendix A) exhibited a high concentration of kaempferol (18.22 ± 2.48 mg/100 g DM), diosmin + hesperidin (169.06 ± 24.34 mg/100 g DM), apigenin (4.59 ± 0.48 mg/100 g DM) and diosmetin (9.71 ± 0.57 mg/100 g DM).

According to the statistical analysis (Appendix A) of the factorial design, it was observed that the factor by-product had a greater influence (*p* < 0.05) on 15 of the 17 polyphenols analyzed, followed by the factor solvent with a significant effect (*p* < 0.05) on 10 polyphenols (gallic acid, catechin, chlorogenic acid, kaempferol, vanillin, diosmin, hesperidin, naringenin, apigenin and diosmetin); the interaction of both factors had a significant influence (*p* < 0.05) in 8 different polyphenols. The variety showed less influence on the response variables (six polyphenols), in terms of interactions. On the other hand, the interaction that presented the least influence (one polyphenol) was the variety with the solvent. Finally, the triple interaction did not influence any polyphenol identified in the extracts obtained from the by-products of both habanero pepper varieties.

### 2.3. Vitamin C

All the extracts obtained with MEOH from both by-products and both habanero pepper varieties presented a lower concentration compared to the rest of the extracts from the experimental design (*p* > 0.05).

The extract derived from NADES-h from the by-product leaf of the Mayapan variety habanero pepper exhibited the highest concentration of vitamin C (50.43 ± 0.00 mg/100 g DM), followed by the extract achieved NADES-cl, showing a concentration of 28.49 ± 0.98 mg/100 g DM (Table 3).

According to the statistical analysis (Appendix A), the main factors with a significant effect (*p* > 0.05) on the vitamin C concentration in extracts are the solvent and the variety; in addition, the interaction between variety and by-product factors had a significant effect (*p* > 0.05).

### 2.4. Vitamin A and E

Extracts of by-products of habanero pepper, leaf and stem of both varieties, extracted with NADES and MeOH solvents, did not report detectable concentrations of vitamin A and E.

### 2.5. Carotenoids

Lutein was identified and quantified in some extracts of the experimental design (Experiment 2, 9, 11 and 12), as shown in Table 3. The highest concentration of lutein (0.29 mg/100 g DM) was found in extracts obtained from the Mayapan variety leaf by-product with MeOH (*p* < 0.05).

Among the extracts obtained with NADES-cl, only those derived from the stem of the Mayapan variety habanero pepper exhibited a low lutein concentration (0.08 mg/100 g DM). Extracts obtained with NADES-h did not exhibit lutein concentration.

β-carotene was not identified in any of the extracts obtained from the experimental design.

### 2.6. Correlation Analysis

Results obtained from correlation analysis of the individual concentrations of polyphenols and vitamin C, as well as the total polyphenol content of the extracts obtained from the by-products of habanero pepper according to the antioxidant capacity determined by DPPH and ABTS, are shown in Table 4 (equations are presented in Appendix A).

A good correlation (r^2^ > 0.7) was found between protocatechuic acid, neohesperidin and vanillin with antioxidant capacity by the DPPH methodology, the vanillin with the best linear fit (r^2^ = 0.8175), in extracts derived from habanero pepper stem. Those polyphenols did not present the same tendency as by the ABTS methodology; however, vitamin C extracted from habanero pepper stem did show a greater correlation with antioxidant capacity (r^2^ = 0.9747) followed by chlorogenic acid (r^2^ = 0.8517).

In relation to leaf, only a correlation between gallic acid and chlorogenic acid with antioxidant capacity by DPPH (r^2^ = 0.7468) and ABTS (r^2^ = 0.7996) was demonstrated, respectively. Additionally, catechin found in the habanero pepper leaf showed a slight correlation with antioxidant capacity in both DPPH (r^2^ = 0.6660) and ABTS (r^2^ = 0.6932).

The total polyphenol content in *Capsicum chinense* only exhibited a slight correlation (r^2^ = 0.6670) in the leaf with the antioxidant capacity by the ABTS methodology.

### 2.7. Pearson’s Correlation

The connection between the results obtained from the total polyphenol content, individual polyphenols and vitamin C from the experimental design, with antioxidant capacity by the ABTS assay and DPPH radical inhibition, was analyzed by Pearson analysis, considering a positive correlation when r ≥ 0.7 or inverse correlation when r ≤ −0.7; the results are presented at Figure 2.

Total polyphenol content (r = −0.72) presents a negative correlation with antioxidant capacity by ABTS methodology, while it does not present a correlation (r = −0.17) with DPPH methodology, but a similar trend is observed. Diosmin + hesperidin (r = 0.8) and apigenin (r = 0.74) present a positive correlation with TCP.

Individual polyphenols such as protocatechuic acid, rutin, quercetin, luteolin and vanillin present a negative correlation with antioxidant capacity (DPPH). Catechin (r = 0.75) and chlorogenic acid (r = 0.8) exhibit a positive correlation with antioxidant capacity by ABTS methodology.

### 2.8. Principal Component Analysis (PCA)

To explore the extract metabolites’ inequalities and similarities obtained from the stem and leaf by-products of the Mayapan and Jaguar habanero pepper varieties using eutectic solvents and methanol, PCA was conducted. Figure 3a shows the combined graph (bi-plot) of the variables (type of solvent) and observations (polyphenols, vitamin C, carotenoids, TPC and antioxidant capacity) from the experimental design. With the first two main components, the data variability is explained more than 90%; principal component 1 (CP1) presents 60.94% of the total dataset variation.

According to the PCA chart (Figure 3a), a similar distance of NADES-cl from NADES-h and MeOH indicates that the first shares characteristics with the last solvents, as well as its effect on the response variables; NADES-h and MeOH exhibited greater distance from each other, implying differences between both solvents. Polyphenols such as cinnamic acid (13), coumaric acid (12), gallic acid (2), diosmetin (10), naringenin (14), neohesperidin (9), apigenin (15), vanillin (11), kaempferol (8), rutin (7), lutein (16) as carotenoid and the antioxidant capacity assessed by ABTS methodology were not associated with any solvent; this cluster was observed near to the center of the plot (Figure 3a).

Protocatechuic acid (3), catechin (4) and quercetin + luteolin (6) established a conglomerate that may be associated more with NADES-h; meanwhile, total polyphenol content (1), diosmin + hesperidin (17) and antioxidant capacity by DPPH are associated to both NADES-cl and MeOH (Figure 3b).

At Figure 3c and d, it can be observed that DPPH and catechin (4) forms a conglomerate with a strong association with the habanero pepper stem. The cluster formed by rutin (7), TPC (1), quercetin + luteolin (6), vitamin C (18), gallic acid (2) and protocatechuic acid (3) also showed a trend in the association with the habanero pepper stem, but they are closer to the plot’s center (low association). Diosmin + hesperidin (17) formed a unique conglomerate with a strong association to habanero pepper’s leaf. Polyphenols such as cinnamic acid (13), coumaric acid (12), gallic acid (2), diosmetin (10), naringenin (14), neohesperidin (9), vanillin (11), kaempferol (8) and the carotenoid lutein (16), identified in extracts, are not associated with any habanero pepper by-products.

Main component analysis of the habanero pepper variety as a factor (Appendix A) highlights a greater association of individual polyphenols (3, 4 and 17), as well as total content of polyphenols with Jaguar variety, while the vitamin C, quercetin + luteolin and antioxidant capacity (DPPH) are associated with the Mayapan variety. As in the solvent and by-product ACPs, several individual polyphenols (13, 14, 2, 4, 8, 9, 10, 11, 14, 15), lutein (16) and antioxidant capacity (ABTS) are not associated with any variety of habanero pepper.

Finally, it can be distinguished that both green solvents, NADES-cl and NADES-h, have a better association and effect on the extraction of various phenolic compounds, thus improving the polyphenol profile of extracts obtained from habanero pepper stem in comparison with extracts obtained by an organic solvent such as methanol.

## 3. Discussion

The highest concentration of total polyphenols (TPC) was found in the extract derived from the Jaguar leaf using MeOH as solvent, however, the rest of the methanolic extracts presented lower concentrations compared to the extracts obtained with NADES-h and NADES-cl. This variability in the TPC may be due to the fact that other different bioactive compounds present in the MeOH solvent extract may have the ability to interfere with the Folin reagent, such as unidentified phenolic compounds and/or metabolites such as vitamin C [24], terpenes, amino acids, proteins or sugars, separated from the food matrix during the extraction process [25,26].

The absence of catechin, a typical polyphenol previously reported in habanero pepper [27], and gallic acid in the leaf and stem extract of habanero pepper Jaguar variety was reported by Chel-Guerrero et al. [3], using methanol (80%) as a solvent by ultrasound-assisted extraction. However, in the present work, the highest concentrations of catechin were observed in the Jaguar variety stem extracts obtained with NADES-h (69.88 ± 1.16 mg catechin/100 g MS) and NADES-cl (59.70 ± 6.04 mg catechin/100 g MS) compared to those obtained with MeOH (0.12 ± 0.00 mg catechin/100 g MS); gallic acid was also extracted, obtaining the highest concentration in the leaf extract (Mayapan variety) and with NADES-cl (15.21 ± 3.71 mg gallic acid/100 g MS). The extraction and presence of these phenolic compounds in extracts where NADES were used was due to their affinity with the hydrogen bonds formed between HBD and HBA [28]. Due to this characteristic, extracts obtained by NADES presented a better polyphenol profile as well as a higher concentration in comparison to MeOH extracts.

In this work, extracts derived from the NADES-h from the leaves of the Mayapan variety had higher concentrations of rutin (28.62 ± 0.58 mg/100 g DM) and quercetin (52.68 ± 2.51 mg/100 g DM) compared to methanolic extracts from Mayapan leaf (rutin: 10.16 ± 1.15 mg/100 g DM; quercetin: 9.00 ± 0.84 mg/100 g DM). These results are similar to those obtained by Zhou et al. [29], who evaluated the phenolic compound extraction from *Morus alba* L. leaf using different eutectic and organic solvents (methanol and water), where the eutectic solvents extracts yielded a higher concentration of rutin and quercetin compared to those obtained with organic solvents. Using a scanning electron microscopy (SEM), they noticed that the cell walls of mulberry leaves were entirely damaged when treated with NADES and partially damaged when treated with methanol. This phenomenon could explain the total release of phenolic compounds and the better dilution of the metabolites in the solvent. It is now known that NADES have a better hydrophilicity than methanol.

The hydrophilicity of NADES is an advantage for the extraction of other metabolites of interest, such as vitamin C, one which has hydrophilic characteristics and, as reported, increases yield extraction when added to water and in the presence of HBA, such as choline [30]. For this reason, higher concentrations of ascorbic acid were observed in both leaf and stem extracts from both habanero pepper varieties, Jaguar and Mayapan, obtained with NADES-cl and NADES-h, compared to those extracts where MeOH was used as a solvent.

In contrast, the NADES’ hydrophilic capacity, which is also increased by the addition of water, reduces the affinity for hydrophobic compounds and, therefore, the ability to extract for example lutein (carotenoid) [31]. This is consistent with the present work, where the low concentration of carotenoids, mainly β-carotene that is absent in the extracts, and a very low or null concentrations of lutein in the extracts of both by-products of habanero pepper obtained by both NADES (NADES-cl and NADES-h) were demonstrated. Therefore, it is expected that extracts with a higher concentration of carotenoids and hydrophobic eutectic solvents must be implemented, such as decanoic acid:Lidocaine (HBD:HBA, 2:1 molar), menthol:Lidocaine (HBD:HBA, 2:1 molar) and thymol:coumarin (HBD:HBA, 1:1 molar), among others [32].

The antioxidant capacity of the extracts obtained with the organic solvent (MeOH) was higher when assessed by the DPPH methodology than in the ABTS methodology, where the extracts obtained with NADES presented the highest antioxidant capacity. This could be due to the individual phenolic profile of the extracts; for example, in the MeOH extracts, diosmin was the main phenolic compound and showed a higher concentration compared to the rest of the phenolic compounds identified in the extract. According to Platzer et al. [33], diosmin has a high number of hydroxyl groups in its chemical structure (more than five). This characteristic allows the neutralization of the DPPH radical (interaction with the phenolic compound), which translates into a high inhibition percentage; the results are the antioxidant capacity of this metabolite. This behavior can be corroborated with the PCA (Figure 3), where this metabolite is associated with the DPPH methodology. Furthermore, this methodology is influenced by using a methanolic medium, since the mechanism, by which the transfer of a proton from the antioxidant to the oxidant is by the DPPH methodology, occurs preferably in alcohol solvents such as methanol and ethanol, while, in the ABTS methodology, this transfer occurs on aqueous solutions [34]. Furthermore, metabolites such as vitamin C with two hydroxyl groups or phenolic compounds with three or four hydroxyl groups have higher reactivity with the oxidizing agent ABTS [35,36]. In the extracts obtained with NADES from the stems and leaves of both varieties of habanero pepper, phenolic compounds, with these hydroxyl groups, were identified, such as gallic acid, protocatechuic acid, luteolin and naringenin, all in higher concentration compared to methanolic extracts. This finding could explain the higher antioxidant capacity of NADES with the ABTS methodology [34]; additionally, a correlation between those metabolites (vitamin C and individual polyphenols) and ABTS can be observed in the linear correlation (Table 4), Pearson correlation (Figure 2) and the principal component analysis (Figure 3).

## 4. Materials and Methods

### 4.1. Chemical Reagents

NADES reagents, choline chloride and D-glucose, were obtained from Sigma Aldrich^®^ (St. Louis, MO, USA) as the standards (gallic acid, protocatechuic acid, chlorogenic acid, coumaric acid, cinnamic acid, vanillin, catechin, apigenin, diosmetin, rutin, kaempferol, quercetin, luteolin, hesperidin, diosmin, neohesperidin, naringenin, Ascorbic acid, lutein and β-carotene) used for the quantitative analysis of the metabolites in the samples.

Methanol, acetonitrile, acetic acid and formic acid were HPLC grade solvents obtained from Sigma Aldrich (Steinheim, Germany).

### 4.2. By-Product Material

The leaves and stems of two varieties of habanero pepper (*Capsicum chinense* Jacq.), Mayapan and Jaguar, obtained from habanero pepper plants cultivated in plastic pots with black soil (*Boox Lu’um*, Mayan name) for a period of 7 months (24 May–6 December 2021) and under greenhouse conditions, were processed at The Centro de Investigación y Asistencia en Tecnología y Diseño del Estado de Jalisco, A.C. (CIATEJ) Subsede Sureste (Latitude N 21°8′1.288″ and Longitude W 89°46′52.26″).

### 4.3. Honey (NADES-h)

Honey harvested on 2 March 2022 was obtained from a local producer. It represents a multifloral honey (*Viguiera dentata* and *Gymnopodium floribundum*) made by *Apis mellifera* bees at Telchaquillo, Yucatán (Latitude N 20°37′38.1″ and Longitude W 89°27′50.8″).

The honey was diluted with distilled water at a proportion of 60:40 (honey:water, *v*/*v*) to reduce viscosity. In order to achieve only the antioxidant capacity of the bioactive compounds obtained by ultrasound-assisted extraction with the help of NADES-h, the antioxidant capacity obtained from honey was subtracted from the antioxidant capacity of NADES-h extracts as well as every analyzed compound (TPC, polyphenols profile, vitamin C and carotenoids).

### 4.4. Drying of Habanero Pepper’s By-Products

Drying was carried out according to Chel-Guerrero et al. [3], with some modifications. The first step was to separate the leaves and stems of the habanero pepper plants; the stems were cut with a knife and both by-products were placed in stainless steel trays inside a FELISA oven (FE-292) for 48 h at 44 °C to obtain a 5% humidity or less. Once the drying process was finished, the by-products were ground with a coffee grinder (Braun^®^, model KSM-2) and the powder obtained was sieved with a particle size mesh of 500 µm (# 35, Fisher Scientific, Boston, MA, USA). Finally, the habanero pepper’s by-product flour was protected with a plastic bag lined with aluminum foil and stored at room temperature until use.

### 4.5. Preparation of NADES

To obtain NADES-cl, the procedure reported by Mansinhos et al. [37] was followed. The components were mixed in a 1:1 molar ratio, with choline chloride being the hydrogen bond acceptor (HBA) and glucose the hydrogen bond donor (HBD), then heated in a water bath at a constant stirring and temperature of 90 °C until a pale yellowish liquid phase was formed. To improve extraction and decrease the viscosity of NADES-cl and NADES-h, they were mixed with distilled water in a 60:40 *v*/*v* (NADES: water) ratio.

### 4.6. Ultrasound-Assisted Extraction of Phenolic Compounds from Habanero Pepper’s By-Products

The phenolic compound extraction procedure, established by Chel-Guerrero et al. [7], was followed, with some modifications. A sample of 0.5 g of by-product flour (leaf or stem) was taken and placed in falcon tube, then, 5 mL of solvent (NADES, honey or Methanol) was added and mixed in a vortex. Next, an ultrasonic bath was applied for 30 min at 42 kHz with BRANSON^®^ equipment (model 351). Subsequently, the samples were centrifuged at 4700 rpm for 30 min at 4 °C and the supernatants were recovered and filtered with a nylon filter (22 µm, pore size); finally, the samples were placed in amber chromatographic vials and settled under refrigeration until use.

### 4.7. Evaluation of Total Polyphenol Content (TPC)

Extracts were evaluated using the Folin–Ciocalteu methodology according to Singleton et al. [38], with some modifications, where 25 µL of the extracted sample was diluted with distilled water (25 µL), then 3 mL of distilled water and 250 µL of Folin’s reagent were added, left to stand for 5 min, and then 750 µL of Sodium Carbonate (NaCO_3_) at 20% and 950 µL of distilled water were added and incubated for 30 min. Lastly, the samples’ signals were read at 765 nm with a Uv-Vis spectrophotometer (JENWAY^®^, model 6700) to determine the absorbance. Results were reported as milligram of gallic acid equivalent per 100 g of dry matter (GAE/100 g DM) according to the calibration curve, reported in Appendix A.

### 4.8. Evaluation of Extracts Antioxidant Capacity

#### 4.8.1. DPPH Methodology

The antioxidant capacity of extracts obtained by ultrasound-assisted extraction using different solvents was evaluated according to Oney-Montalvo et al. [1], where 3.3 mg of DPPH were weighed, adding methanol until a volume of 100 mL was reached, then the solution was adjusted with methanol to an 0.700 ± 0.002 absorbance at 515 nm with a THERMO SCIENTIFIC^®^ Uv-Vis spectrophotometer (Genesys 140 model).

A sample of 100 µL was mixed with 3.9 mL of DPPH (adjusted) and incubated for 30 min; finally, the absorbance was read and the antioxidant capacity was reported as the percentage of inhibition, according to Equation (1).
% DPPH Inhibited = 100 − [(Abs extract − Abs DPPH adjusted/Abs DPPH adjusted) × 100](1)

#### 4.8.2. ABTS Methodology

Habanero pepper by-products’ extracts (*Capsicum chinense* Jacq.) were also evaluated by the ABTS methodology to obtain the antioxidant capacity in mM Trolox equivalents. According to Re et al. [39], the ABTS (STA) working solution was prepared by adding 25 µL of hydrogen peroxide to the ABTS substrate solution and the working myoglobin solution (SMT), making up to 1 mL with the working buffer solution.

For each extract, 20 µL was added to 40 µL of SMT, then 300 µL of STA was added, shaken and allowed to stand for 5 min; finally, 200 µL of stop solution was added, incubating for 60 min to finally read at 405 nm.

### 4.9. Quantification and Identification of Individual Phenolic Compounds

Determination of the polyphenols was carried out with a diode array detector (DAD) coupled to an UPLC Acquity H Class equipment (Waters, Milford, MA, USA), where an Acquity UPLC HSS C18 column was used.

A calibration curve was made with 17 polyphenol standards (Sigma-Aldrich^®^). To develop the curve, a stock solution was prepared with a concentration of 1 mg/mL with the following polyphenols: gallic acid, protocatechuic acid, chlorogenic acid, coumaric acid, cinnamic acid, catechin, rutin, kamepferol, quercetin, luteolin, vanillin, diosmin, hesperidin, neohesperidin, naringenin, apigenin and diosmetin. The resulting chromatograms can be seen at Appendix A.

Both the calibration curve and the extracts were analyzed with a column temperature of 45 °C and an injection volume of 2 µL. A wavelength of 280 nm was established and 0.2% acetic acid was used as solvent A and acetonitrile with 0.1% acetic acid as solvent B. The elution gradient was as follows: 0–10 min from 99% A to 70% A; 10–12 min 70% A; 12–15 min from 70% A to 99% A. Every injection lasted 15 min.

### 4.10. Determination of Vitamins A, E and C

Determination of vitamins A and E was carried out according to Rodríguez-Buenfil et al. [8], where a flow rate of 0.5 mL/min was used, injecting the mobile phase (2 µL volume) acetonitrile: methanol (50:50). Measurements were made at 290 nm wavelength with a DAD. The retention times of the calibration curve of α-tocopherol and retinol (1.5 to 75 µg/mL) were compared with the samples for identification.

Vitamin C was determined by a mobile phase of 0.1% formic acid, injected in a 2 µL volume at a flow rate of 0.25 mL; the column was maintained at a temperature of 27 °C and measures were made with a DAD at a wavelength of 244 nm, according to Rodríguez-Buenfil et al. [8]. A calibration curve of ascorbic acid (0.5 to 5 µg/mL) was prepared prior to injection of the samples to compare retention times.

### 4.11. Determination of Carotenoids

A calibration curve was prepared for determination of carotenoids with β-carotene and lutein standards; the retention times were compared with those obtained in the samples. The injection conditions were carried out according to Rodríguez-Buenfil et al. [8] using acetonitrile:methanol (70:30) as the mobile phase at a flow rate of 0.5 mL/min. The reading was made with a DAD (diode array detector) at a wavelength of 475 nm.

### 4.12. Statistical Analysis

Experiments were performed randomly and duplicated for each extract from the experimental design. Data presented are reported as means ± standard deviations. The analysis of linear correlation was carried out between the concentrations of by-products’ total and individual polyphenols, extracted with the antioxidant capacity obtained by the DPPH and ABTS methodology; it was carried out with a Pearson’s correlation coefficient I as well as a principal component analysis (PCA). Data analysis was performed using the statistical software Statgraphics Centurion XVII.II-X64 (Statgraphics Technologies Inc. Virgin, UT, USA), XLSTAT 2021.2.2 (Addison, Paris, France) and with R 4.0.3 (The R Foundation for Statistical Computing, Vienna, Austria).

## 5. Conclusions

The results indicate that the type of solvent’s and by-product’s individual contributions and the interaction of both presented an effect on the total content of polyphenols, the antioxidant capacity and on almost all the individual polyphenols of the analyzed profile. In relation to the concentration of vitamin C in the extracts, the type of solvent and the variety of pepper had an effect individually, as well as the interaction between this last factor and the by-product factor. NADES showed a better profile of polyphenols extracted from the leaf and the stem of both varieties of habanero pepper compared to the extracts obtained with MeOH, where NADES presented high concentrations of gallic acid, protocatechuic acid, luteolin and naringenin, metabolites that participate in the antioxidant capacity of the extracts, being higher in comparison with extracts obtained with organic solvent. The highest concentration of vitamin C was found in the Mayapan variety of habanero pepper leaf extract obtained with NADES-h. The association of individual polyphenols with a type of solvent, by-product or variety of habanero pepper was also demonstrated. This allows us to conclude that the use of NADES for the extraction of phenolic compounds from by-products (leaves and stems) of habanero pepper (*Capsicum chinense* Jacq.) is a promising strategy for the revaluation of the waste derived from this industry and to help the decrease in the use of organic solvents. This knowledge will allow, according to the needs of bioactive compounds, proper selection of the raw material and the reagents to obtain high concentrations of specific individual phenolic compounds without losing their antioxidant capacity.

## Figures and Tables

**Figure 1 plants-11-02060-f001:**
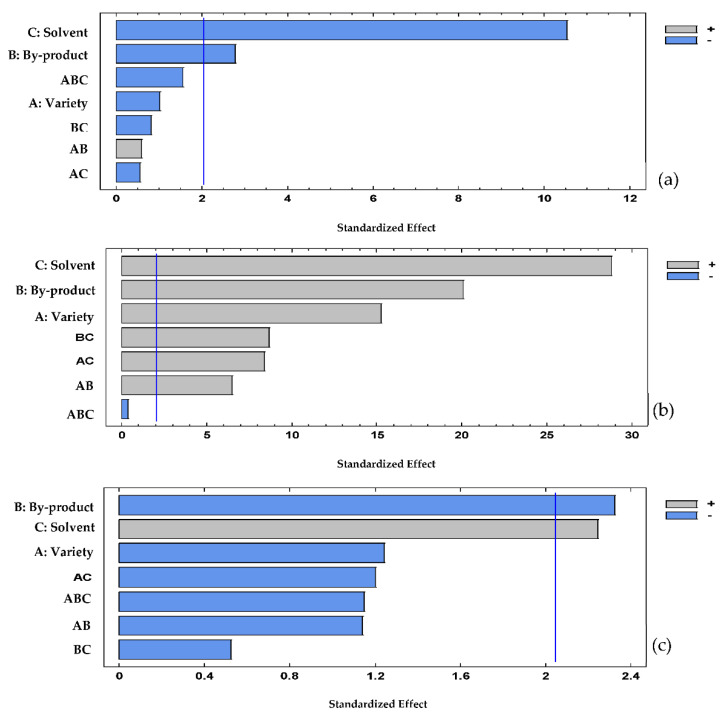
Pareto chart: (**a**) total polyphenol content, (**b**) DPPH (**c**) and ABTS.

**Figure 2 plants-11-02060-f002:**
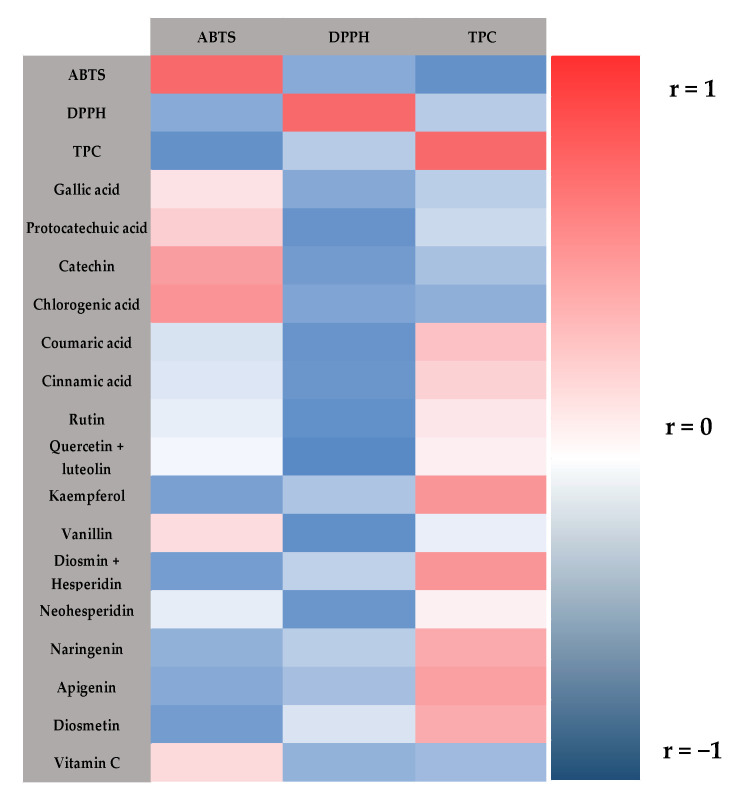
Heat map of the Pearson’s correlation of total polyphenol content, individual phenolic compounds and vitamin C with antioxidant capacity; TPC: Total polyphenol content.

**Figure 3 plants-11-02060-f003:**
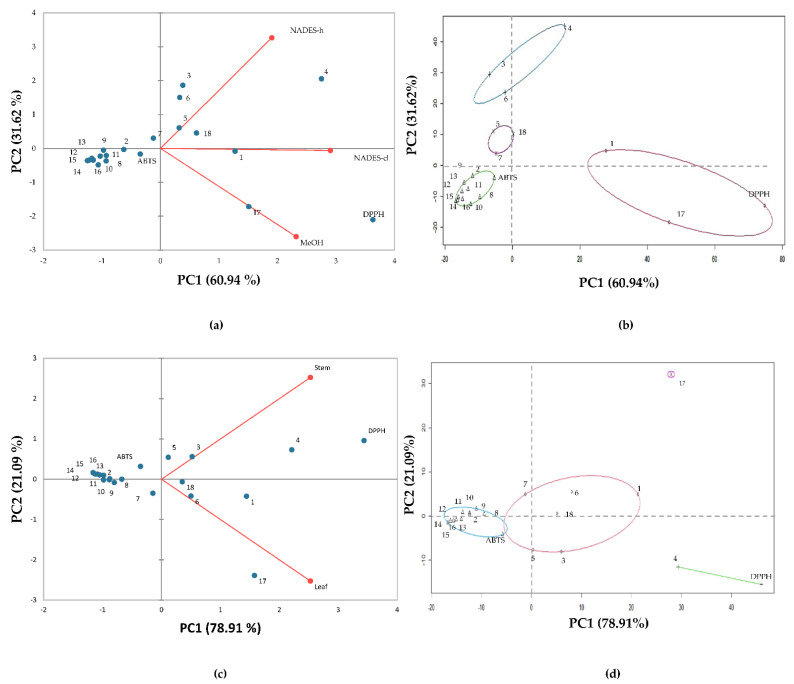
Analysis of total polyphenol content, polyphenol profile and antioxidant capacity: principal component analysis (PCA) depending on solvents (**a**) and by-product (**c**) used, and cluster of k means depending on solvents (**b**) and by-product (**d**) used. Numeration: 1 = total polyphenol content; 2 = gallic acid; 3 = protocatechuic acid; 4 = catechin; 5 = chlorogenic acid; 6 = quercetin + luteolin; 7 = rutin; 8 = kaempferol; 9 = neohesperidin; 10 = diosmetin; 11 = vanillin; 12 = coumaric acid; 13 = cinnamic acid; 14 = naringenin; 15 = apigenin; 16 = lutein, 17 = diosmin + hesperidin; 18 = vitamin C.

**Table 1 plants-11-02060-t001:** Total polyphenol content and antioxidant capacity of extracts obtained by UAE from by-products of habanero pepper (*Capsicum chinense* Jacq.).

# Exp	Factors			Variable Response		
	Variety	By-Product	Solvent *	TPC (mg GAE/100 g DM)	DPPH (% Inhibition)	ABTS (eq mg Trolox/g DM)
1	Mayapan	Stem	NADES-cl	9.18 ± 0.29 ^b^	54.21 ± 0.91 ^e^	**9.55 ± 0.02 ^l^**
2	Jaguar	Stem	NADES-cl	7.60 ± 0.56 ^a^	53.88 ± 1.11 ^e^	8.13 ± 0.05 ^h^
3	Mayapan	Leaf	NADES-cl	11.00 ± 0.54 ^c^	36.99 ± 1.30 ^d^	7.44 ± 0.08 ^f^
4	Jaguar	Leaf	NADES-cl	**22.38 ± 0.07 ^f^**	28.48 ± 1.95 ^c^	8.63 ± 0.03 ^j^
5	Mayapan	Stem	NADES-h **	18.08 ± 0.03 ^d^	14.74 ± 2.81 ^b^	8.75 ± 0.01 ^k^
6	Jaguar	Stem	NADES-h **	24.60 ± 0.81 ^g^	14.08 ± 5.69 ^b^	8.45 ± 0.03 ^i^
7	Mayapan	Leaf	NADES-h **	28.12 ± 0.04 ^h^	7.09 ± 2.28 ^a^	7.85 ± 0.01 ^g^
8	Jaguar	Leaf	NADES-h **	**39.31 ± 0.09 ^k^**	12.55 ± 2.73 ^ab^	7.16 ± 0.01 ^e^
9	Mayapan	Stem	MeOH	19.37 ± 0.03 ^e^	**82.83 ± 4.12 ^f^**	3.94 ± 0.02 ^d^
10	Jaguar	Stem	MeOH	33.90 ± 6.83 ^i^	**81.36 ± 4.10 ^f^**	3.58 ± 0.01 ^c^
11	Mayapan	Leaf	MeOH	37.75 ± 6.83 ^j^	78.32 ± 1.22 ^f^	2.62 ± 0.01 ^b^
12	Jaguar	Leaf	MeOH	**63.80 ± 0.00 ^l^**	77.84 ± 1.10 ^f^	1.58 ± 0.01 ^a^
Control	-	-	Honey ***	0.67 ± 0.01	51.88 ± 2.27	0.44 ± 0.00

**Note:** Exp = Experiment; NADES-cl = Natural deep eutectic solvent (choline chloride: glucose, 1:1 molar); NADES-h = Natural deep eutectic solvent (honey); MeOH = Methanol; TPC = Total polyphenol content; * Proportion of NADES-cl: water (60:40); NADES-h: water (60:40); MeOH: water (80:20); ** values obtained are the result of the extract analysis minus the honey values analyzed; *** Proportion of Honey:water (60:40). Different lowercase letters on each column show a statistically significant difference; Values are means ± SD (*n* = 3).

**Table 2 plants-11-02060-t002:** Polyphenol profile from habanero pepper (*Capsicum chinense* Jacq.) by-products extracts.

# Exp	Factors			Polyphenol Profile (mg/100 g DM)					
	Variety	By-Product	Solvent *	Gallic Acid	Protocatechuic Acid	Catechin	Chlorogenic Acid	Coumaric Acid	Cinnamic Acid
1	Mayapan	Stem	NADES-cl	2.82 ± 0.06 ^b^	2.91 ± 0.04 ^a^	50.20 ± 0.23 ^d^	15.74 ± 0.38 ^d^	0.00 ± 0.00 ^a^	2.06 ± 0.06 ^abc^
2	Jaguar	Stem	NADES-cl	0.82 ± 0.08 ^ab^	16.82 ± 12.31 ^b^	59.70 ± 6.04 ^e^	27.84 ± 2.28 ^f^	0.02 ± 0.03 ^a^	1.14 ± 0.61 ^ab^
3	Mayapan	Leaf	NADES-cl	**15.21 ± 3.71 ^d^**	1.36 ± 0.96 ^a^	9.64 ± 11.53 ^b^	7.57 ± 0.10 ^c^	0.00 ± 0.00 ^a^	2.58 ± 0.05 ^bc^
4	Jaguar	Leaf	NADES-cl	8.45 ± 0.74 ^c^	2.73 ± 0.17 ^a^	39.72 ± 0.81 ^c^	17.35 ± 0.09 ^e^	1.68 ± 0.07 ^e^	4.97 ± 0.83 ^d^
5	Mayapan	Stem	NADES-h **	8.21 ± 0.15 ^c^	36.79 ± 12.31 ^c^	39.80 ± 6.04 ^c^	18.73 ± 2.28 ^e^	0.56 ± 0.03 ^b^	2.30 ± 0.61 ^bc^
6	Jaguar	Stem	NADES-h **	6.42 ± 0.02 ^c^	**59.63 ± 0.13 ^d^**	**69.88 ± 1.16 ^f^**	**30.29 ± 0.34 ^g^**	1.45 ± 0.00 ^d^	2.03 ± 0.03 ^abc^
7	Mayapan	Leaf	NADES-h **	0.66 ± 0.02 ^ab^	37.91 ± 0.13 ^c^	46.94 ± 0.26 ^cd^	10.58 ± 0.70	1.04 ± 0.12 ^c^	4.88 ± 0.05 ^d^
8	Jaguar	Leaf	NADES-h **	13.10 ± 0.02 ^d^	39.78 ± 0.08 ^c^	52.39 ± 0.34 ^de^	18.78 ± 0.23 ^e^	**2.77 ± 0.07 ^f^**	**7.20 ± 0.41 ^e^**
9	Mayapan	Stem	MeOH	0.00 ± 0.00 ^a^	0.00 ± 0.00 ^a^	0.14 ± 0.00 ^a^	0.00 ± 0.00 ^a^	0.00 ± 0.00 ^a^	0.00 ± 0.00 ^a^
10	Jaguar	Stem	MeOH	0.00 ± 0.00 ^a^	0.00 ± 0.00 ^a^	0.12 ± 0.00 ^a^	0.00 ± 0.00 ^a^	0.00 ± 0.00 ^a^	0.00 ± 0.00 ^a^
11	Mayapan	Leaf	MeOH	2.31 ± 0.05 ^b^	0.00 ± 0.00 ^a^	12.83 ± 0.98 ^b^	0.98 ± 0.11 ^ab^	0.41 ± 0.00 ^b^	3.37 ± 0.14 ^cd^
12	Jaguar	Leaf	MeOH	1.92 ± 0.22 ^ab^	0.00 ± 0.00 ^a^	16.62 ± 2.58 ^b^	1.77 ± 0.27 ^b^	1.65 ± 0.32 ^de^	4.91 ± 0.81 ^d^
Control	-	-	Honey ***	0.25 ± 0.00	0.27 ± 0.02	0.00 ± 0.00	0.00 ± 0.00	0.00 ± 0.00	0.00 ± 0.00
**# Exp**	**Polyphenol Profile** **(mg/100 g DM)**								
	**Rutin**	**Quercetin + Luteolin**	**Kaempferol**	**Vanillin**	**Diosmin + Hesperidin**	**Neohesperidin**	**Naringenin**	**Apigenin**	**Diosmetin**
1	2.71 ± 0.14 ^ab^	0.72 ± 0.05 ^a^	0.00 ± 0.00 ^a^	1.41 ± 0.01 ^b^	1.93 ± 0.52 ^a^	0.66 ± 0.53 ^a^	0.00 ± 0.00 ^a^	0.00 ± 0.00 ^a^	0.00 ± 0.00 ^a^
2	6.36 ± 2.07 ^bcd^	0.63 ± 0.02 ^a^	0.00 ± 0.00 ^a^	1.70 ± 0.06 ^bc^	0.00 ± 0.00 ^a^	0.00 ± 0.00 ^a^	0.00 ± 0.00 ^a^	0.00 ± 0.00 ^a^	0.00 ± 0.00 ^a^
3	7.40 ± 2.36 ^cd^	2.99 ± 0.10 ^a^	0.46 ± 0.02 ^ab^	1.54 ± 0.05 ^b^	2.73 ± 0.67 ^a^	1.88 ± 1.33 ^a^	0.00 ± 0.00 ^a^	0.00 ± 0.00 ^a^	0.00 ± 0.00 ^a^
4	16.82 ± 4.77 ^ef^	10.07 ± 0.85 ^b^	1.76 ± 0.10 ^bc^	4.14 ± 0.55 ^f^	59.64 ± 0.98 ^b^	1.16 ± 0.09 ^a^	0.00 ± 0.00 ^a^	1.27 ± 0.01 ^b^	5.25 ± 0.34 ^b^
5	4.92 ± 2.07 ^bc^	28.30 ± 0.02 ^d^	3.11 ± 0.00 ^c^	1.38 ± 0.06 ^b^	2.03 ± 0.00 ^a^	4.94 ± 0.00 ^a^	0.00 ± 0.00 ^a^	1.52 ± 0.00 ^b^	0.00 ± 0.00 ^a^
6	14.21 ± 0.13 ^e^	30.89 ± 0.11 ^d^	3.37 ± 0.02 ^c^	1.93 ± 0.02 ^cd^	6.63 ± 0.06 ^a^	0.00 ± 0.00 ^a^	0.00 ± 0.00 ^a^	1.58 ± 0.01 ^b^	0.00 ± 0.00 ^a^
7	**28.62 ± 0.58 ^h^**	**52.68 ± 2.51 ^e^**	7.82 ± 0.12 ^d^	2.78 ± 0.03 ^e^	14.46 ± 0.08 ^a^	**10.60 ± 0.29 ^b^**	0.00 ± 0.00 ^a^	1.95 ± 0.01 ^c^	0.00 ± 0.00 ^a^
8	18.93 ± 0.23 ^g^	49.38 ± 5.60 ^e^	0.55 ± 0.35 ^ab^	**5.31 ± 0.19 ^g^**	11.63 ± 1.30 ^a^	**10.94 ± 8.17 ^b^**	**2.06 ± 0.01 ^c^**	0.00 ± 0.00 ^a^	0.00 ± 0.00 ^a^
9	0.00 ± 0.00 ^a^	0.00 ± 0.00 ^a^	0.00 ± 0.00 ^a^	0.00 ± 0.00 ^a^	0.00 ± 0.00 ^a^	0.00 ± 0.00 ^a^	0.00 ± 0.00 ^a^	0.00 ± 0.00 ^a^	0.00 ± 0.00 ^a^
10	0.00 ± 0.00 ^a^	0.00 ± 0.00 ^a^	0.00 ± 0.00 ^a^	0.00 ± 0.00 ^a^	0.00 ± 0.00 ^a^	0.00 ± 0.00 ^a^	0.00 ± 0.00 ^a^	0.00 ± 0.00 ^a^	0.00 ± 0.00 ^a^
11	10.16 ± 1.15 ^d^	9.00 ± 0.84 ^b^	10.15 ± 1.52 ^e^	0.00 ± 0.00 ^a^	78.13 ± 1.57 ^c^	0.55 ± 0.05 ^a^	1.21 ± 0.18 ^b^	3.02 ± 0.35 ^d^	8.15 ± 0.22 ^c^
12	15.17 ± 2.78 ^ef^	18.38 ± 3.37 ^c^	**18.22 ± 2.48 ^f^**	2.10 ± 0.23 ^d^	**169.06 ± 24.34 ^d^**	4.00 ± 0.63 ^a^	1.14 ± 0.09 ^b^	**4.59 ± 0.48 ^e^**	**9.71 ± 0.57 ^d^**
Control	0.02 ± 0.00	2.71 ± 0.00	0.03 ± 0.01	0.03 ± 0.00	0.59 ± 0.00	0.00 ± 0.00	0.06 ± 0.00	0.15 ± 0.00	0.00 ± 0.00

**Note**: Exp = Experiment; NADES-cl: = Natural deep eutectic solvent (Choline Chloride: Glucose, 1:1 molar); NADES-h = Natural deep eutectic solvent (Honey); MeOH = Methanol; TPC = Total polyphenol content; * Proportion of NADES-cl:water (60:40); NADES-h: water (60:40); MeOH: water (80:20); ** values obtained are the result of the extract analysis minus the honey values analyzed; *** Proportion of honey:water (60:40). Different Lowercase letters on each column show a statistically significant difference; Values are means ± SD (*n* = 3).

**Table 3 plants-11-02060-t003:** Vitamin C and lutein content of extracts obtained by ultrasound from habanero pepper by-products (*Capsicum chinense* Jacq.).

# Exp	Factors			Response Variable	
	Variety	By-Product	Solvent *	Vitamin C (mg/100 g DM)	Lutein (mg/100 g DM)
1	Mayapan	Stem	NADES-cl	16.89 ± 0.09 ^d^	0.00 ± 0.00 ^a^
2	Jaguar	Stem	NADES-cl	14.87 ± 0.08 ^c^	0.08 ± 0.00 ^c^
3	Mayapan	Leaf	NADES-cl	28.49 ± 0.98 ^f^	0.00 ± 0.00 ^a^
4	Jaguar	Leaf	NADES-cl	18.89 ± 0.02 ^e^	0.00 ± 0.00 ^a^
5	Mayapan	Stem	NADES-h **	13.83 ± 1.95 ^c^	0.00 ± 0.00 ^a^
6	Jaguar	Stem	NADES-h **	14.95 ± 1.07 ^c^	0.00 ± 0.00 ^a^
7	Mayapan	Leaf	NADES-h **	**50.43 ± 0.00 ^g^**	0.00 ± 0.00 ^a^
8	Jaguar	Leaf	NADES-h **	0.00 ± 0.00 ^a^	0.00 ± 0.00 ^a^
9	Mayapan	Stem	MeOH	5.67 ± 0.03 ^b^	0.02 ± 0.00 ^b^
10	Jaguar	Stem	MeOH	5.86 ± 0.01 ^b^	0.00 ± 0.00 ^a^
11	Mayapan	Leaf	MeOH	6.17 ± 0.07 ^b^	0.20 ± 0.00 ^d^
12	Jaguar	Leaf	MeOH	6.63 ± 0.02 ^b^	**0.29 ± 0.00 ^e^**
Control	-	-	Honey ***	0.90 ± 0.02	0.00 ± 0.00

**Note:** Exp = Experiment; NADES-cl = Natural deep eutectic solvent (Choline Chloride: Glucose, 1:1 molar); NADES-h = Natural deep eutectic solvent (Honey); MeOH = Methanol, TPC = Total polyphenol content; * Proportion of NADES-cl:water (60:40), NADES-h:water (60:40) MeOH:water (80:20); ** values obtained are the result of the extract analysis minus the honey values analyzed; *** Proportion of honey:water (60:40). Different Lowercase letters on each column show a statistically significant difference; Values are means ± SD (*n* = 3).

**Table 4 plants-11-02060-t004:** Metabolites’ correlation with antioxidant capacity analyzed by DPPH and ABTS methodology in stems and leaves of habanero pepper (*Capsicum chinense* Jacq.).

Polyphenols	DPPH		ABTS	
	Stem	Leaf	Stem	Leaf
	r^2^	r^2^	r^2^	r^2^
gallic acid	0.1136	**0.7468**	0.1174	0.2587
protocatechuic acid	**0.7552**	0.6137	0.2453	0.3129
catechin	0.4431	**0.6660**	0.6143	**0.6932**
chlorogenic acid	0.5399	0.5805	**0.8517**	**0.7996**
coumaric acid	0.4338	0.4082	0.0138	0.0160
cinnamic acid	0.3673	0.2257	0.0406	0.0024
rutin	0.3529	0.4575	0.0114	0.0751
quercetin + luteolin	0.6258	0.4948	0.0890	0.0333
kaempferol	0.0103	0.1726	0.1754	0.4843
vanillin	**0.8175**	0.5249	0.5918	0.1912
diosmin + hesperidin	0.1561	0.1562	0.4659	0.4004
neohesperidin	**0.7345**	0.1266	0.1724	0.0089
naringenin	0.2533	0.0442	0.5176	0.0712
apigenin	0.0139	0.0910	0.1509	0.3206
diosmetin	0.2533	0.1391	0.5176	0.2924
TPC	0.0206	0.1183	0.4507	**0.6670**
vitamin C	0.5163	0.2627	**0.9747**	0.2766

**Note**: TPC = Total polyphenol content.

## Data Availability

Not applicable.

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
