# Peer review of "Antioxidant Capacity, Vitamin C and Polyphenol Profile Evaluation of a Capsicum chinense By-Product Extract Obtained by Ultrasound Using Eutectic Solvent"

_plants, 2022, doi:10.3390/plants11152060_

Round 1

Reviewer 1 Report

I find the manuscript interesting with lot of valuable data and without heavy weaknesses.

Only minor remarks:

Consider changing “by-products” into “leaves and stems” – it will be more precise. Or?

Unify in the text use “Habanero pepper” (like in abstract) and Habanero chili (like in the conclusions)

Abstract 1st sentence: too long thus vague

Lines 100 – 103. Maybe the idea proposed in those lines is better for the whole text, title, new findings etc. It is about new environmentally friendly technology.

Supplementary Table S1: Provide n = ?

Reviewer 2 Report

The manuscript by Avilés-Betanzos et al. is an interesting and innovative application of deep eutectic solvents to extract potential bioactive components in Habanero pepper by-products and evaluate their in vitro antioxidant properties. Nevertheless I have some comments on the manuscript.

-       Table 1: what is the meaning of the superscript letters next to the values in the last three columns? How do the authors explain the higher value of % of DPPH inhibition for honey alone compared to the extracts NADES-h?

-        Figures reporting chromatograms should be reported with the same dimension in terms of x axis, otherwise they are difficult to be compared. Moreover, for the same reason please assign a unique number to each compound (i.e. 1 should be always gallic acid) and provide a number to each peak in all the chromatograms.

-        Why didn’t the authors performed a correlation analysis between Vitamin C and DPPH/ABTS?

-        Lines 329-332: Please explain also in the text Figure 3a to what is referred to.

-        Lines 333-336: from image 3b it seems that the conglomerate is composed of compounds 3,4 and 6. Moreover the authors speak about total polyphenol content which is not present in Figure 3b. Please explain better.

-        Considering that the authors performed an HPLC-UV quantitative analysis they can calculate the total polyphenol content also in this way, removing the possible interference of the Folin-Ciocalteu method and reaching more reliable results. For the peaks not identified they can use the calibration curve of one of the  compound for which they had a standard (e.g. gallic acid).

-        Par. 4.9: please add more information about the % of mobile phase A and B. Was the analysis performed in isocratic conditions? How long is the analysis (minutes)? Which is the concentration range for each compound?

Minor comments:

-        Figure 1c: the legend on the y axis is not completely readable (C: solvent). Please correct.

-        Table 2: “cumaric acid” should be “coumaric acid”. “Vainillin” should be “vanillin”. Please check this last all along the manuscript (also in the tables) because it is often written unproperly.

-        Compound names in the text should all be written with the first letter lowercase.

-        Line 276: “luteolin” should be “lutein”.

-        Line 326: “CP1” should be “PC1”.

-        Lines 342-349: please add the compound number near to each compound as in the above paragraphs.

-        Line 440: “chlorine chloride” should be “choline chloride”.

-        Line 459: what is Aa?

Please revise the English.

Reviewer 3 Report

The manuscript “Antioxidant capacity, vitamin C and polyphenol profile evaluation of a Capsicum chinense by-product extract obtained by ultrasound using eutectic solvent “ is devoted to the investigation of the antioxidant capacity, total polyphenol content (TPC), the phenolic profile and vitamin C content in extracts obtained from by-products (stems and leaves) of two varieties (Mayapan and Jaguar) of habanero pepper, by ultrasound-assisted extraction (UAE) using natural deep eutectic solvents (NADES). The results showed that NADES leads to extracts with significant higher TPC, higher concentrations of individual polyphenols (gallic acid, protocatechuic acid, chlorogenic acid, cinnamic acid, coumaric acid), vitamin C, and, finally, higher antioxidant capacity (9.55 ± 0.02 eq mg Trolox/g DM) than UAE extraction performed with methanol as solvent. The authers cariied out a comparative study on how the type of solvents influences the content of polyphenolic compounds and antioxidant activity in extracts. The manuscript contains a novel idea of using environmentally friendly NADES for extraction of polyphenolic compounds from Capsicum chinense. The experiment is designed correctly. The manuscript is written in good language.

Reviewer 4 Report

The paper is very good and well written, congratulation!

some minor adjusments:

row 79-ancient instead of antient

row 115- choline instead of chlorine

rows 420-421- please rephrase!

row 440-choline instead of chlorine

497- the calibration curve for TPC assay should be added

Round 2

Reviewer 2 Report

The manuscript greatly improved after the modifications. It is now suitable for publication.